# Investigating social development inequality among steel industry workers in Pakistan: A contribution to social development policies

**Shahid Karim** [1,2]*, **Kong Xiang**[1], **Abdul Hameed**[3,4]

**1** The Center for Modern Chinese City Studies, School of Urban and Regional Science, East China Normal University, Shanghai, China, **2** Department of Geography, Government College University, Lahore, Pakistan, **3** Federal Urdu University of Arts, Science and Technology, Islamabad, Pakistan, **4** Innovative Development Strategies (Pvt.) Ltd, Islamabad, Pakistan

\* shahidkarim94@gmail.com

**Data Availability Statement:** All relevant data are within the paper and its Supporting Information files.

## Abstract

Social development of workers has always been a major concern in history. This study, therefore focuses on social development inequalities among steel industry workers in one development zone (Badami Bagh area and along Sheikhupura road of Lahore) of Pakistan. A total of 225 workers were interviewed randomly following the stratified random sampling technique. Principal Component Analysis (PCA) technique was used to construct the socio-economic score (SES) index. Five categories of socioeconomic index were measured using multinomial logistic regression model. A correlation matrix was calculated for PCA. Results revealed that type/size of industry has negative relation while skill level has positive relation with SES. Job status and transport facility did not reflect a significant impact to SES of workers. Correlation matrix depicted that age, experience and medical treatment have positive relation while formal education, number of dependents and distance to job showed a negative trend in relation with SES.

## 1. Introduction

In contemporary world rapid growth and industrialization are driving a debate about workers' well-being and social development [1]. Companies and organizations are under pressure to demonstrate that they are acting in a socially responsible manner [2–5]. However, the mechanism to monitor the wellbeing of workers is largely missing in factories in developing countries. Third-party social audits are the most common type of self-regulation among global businesses manufacturing in developing countries, but these are unable to provide a clear understanding of the needs of the workers as they only consider the visible, physical status of a factory at a single point in time while social development is an ongoing process [6–9]. Despite the fact that there is a large body of research on workplace happiness in developed countries, the concepts are yet to be implemented in developing countries [10].

In general, social development of factory workers refers to improving the well-being of every worker in the industry so they can reach the optimum level of their productivity. Investing in people is a key component of social and human development. It necessitates the removal

**Funding:** This study was conducted with the financial support of The Center for Modern Chinese City Studies, School of Urban and Regional Science, East China Normal University, Shanghai, under Chinese Government Scholarship program. The corresponding author received this academic scholarship to conduct Ph.D. in Human Geography. Innovative Development Strategies provided funding in the form of salary to Abdul Hameed. The funders and authors' affiliations did not have any role in study design, data collection and analysis, decision to publish, or preparation of the manuscript. The specific roles of these authors are articulated in the 'author contributions' section.

**Competing interests:** Dr. Abdul Hameed was a paid employee of "Innovative Development Strategies" during the course of this study. This does not alter our adherence to PLOS ONE policies on sharing data and materials. There are no patents, products in development or marketed products to declare.

of barriers so that all individuals can confidently and dignifiedly pursue their dreams. It is about refusing to accept the fact that poor people will always be poor. Despite the fact that worker's wellbeing and their development is important for their and factory level prodcutvity, factories are providing inadequate safety measures at workplace, and workers frequrnlty face challenges such as higher number of work hours, low wages, management abuses, inhumane working conditions and occupational accidents [11] leading to social injustice and violation of their rights [12, 13]. Preceding studies reported how social order shifts, the types of issues and problems people and their leadership need to tackle [14]. Social development of workers has always been a major concern in the history. Primitive societies even agricultural societies considered and organized social rights of workers in relation to their work. However, modern production structures are, complicated and are mainly controlled by the prodcuers with inadequate regulatrory mechanism for wellbeing of workers [15]. It needs to be recognized that a well-clad, comfortable and satisfied workforceis imperative for productivity and sustainable growth [16]. Parity across workers'categories is also important to improve their satisfaction level and prodcutvity [17]. The industry owners must therefore focus on the growth of employees in the industry. Sustainable income should be given to preserve their families. The working climate, good industrial relations, social services etc. have to be expanded to employees [18]. Social development of workers is not only linked to the cash they earn from their work. There are many activities that create value but are unpaid, such as those that take place in the household [14]. Therefore, socio-economic status should also be associated with other variables in workers lives. For example, two individuals could make the same income, but because of caring obligations and different living conditions, they could have different expenses. If one maintains his own house and the other pays a month's ' mortgage, even people who look after the same amount can face a different domestic situation. Furthermore, there could be extra revenue for certain employees, including rental income and property income that delivers their pay. In order to increase the standard of living of the family, other family members also need salary-earning employment and not just main workers. Based upon the above-mentioned examples, determining the socio-economic status is quite a difficult and complex task [15].

Steel industry remains at the heart of global development. It is regarded the backbone of contemporary development. Considering its usefulness in all industrial procedures, including infrastructure, building sector, automobiles, transport and home appliances, it has a direct relationship with industrial developments. An advanced steel industry is very important for the general economic development of a nation. Steel use is anticipated to increase at an early stage of economic development quicker because enormous amounts of metal are needed for construction of the fundamental infrastructure, including bridges, dams, railways and power generation, distribution and transmission projects, etc. [19]. In most countries, since the end of 2015, the steel market conditions have gradually changed. Market information published in 2018 shows that after reaching 1587.4 million metric tons (mmt) in 2017, the market for steel continues to grow, above the previous peak achieved in 2014 (1545.8 mm). Relative to 2016, it reflects a 4.7% jump. In the first three months of 2018, international exportation of steel fell by nearly 9% annually (y-o-y). The downturn ranged dramatically across countries, with the People's Republic of China dipping significantly by 27%, India decreasing by 35%, the European Union declining by 1%, the Japanese declining by 4%, and USA slipping by 3% [20]. On average, crude steel production has been increased from 189 million tons in 1950 to 1808 million tons worldwide in 2018. In 2017, the steel industry generated US$500 billion value added and a further US$1.2 trillion through its global supply chain. On the employment side the steel industry employs more than 6 million people and that for every 2 jobs in the steel sector, 13 more jobs are supported through its supply chain, in total around 40 million jobs globally [21].

Pakistan is a developing country which is making a tremendous progress in every field of life. The demand and production of steel faced fluctuating trend during recent past. Steel industry witnessed a double-digit decline of 11.0 percent during Jul-Mar FY19 compared to remarkable increase of 27.5 percent during the same period last years [22]. At 29.4kg / capita, which is well below the world average of roughly 233kg / capita, Pakistan is still among the smallest per capita customers of steel, and suggests the huge potential for development in national steel production. Pakistan steel sector was once dominated by the Pakistan Steel Mills Corporation Limited (PSMCL). Approximately there are 600 players in steel industry. These are tiny investors having small production units which are not working in compliance with the standards of working environment. Due to obsolete and inefficient production techniques, their end products are not cost effective, so they cannot compete the cheap products of neighboring countries like India and China [23].

### 1.1. Study objectives and significance

The major objective of the proposed study is to assess the social development inequalities among steel industry workers in Lahore region of Pakistan. The specific objectives of the study are: (a). To find out the socio-economic characteristics of the steelworkers. (b). To evaluate the social and economic condition under which the workers work. (c). To detect the ways and means to increase the efficiency of steel workers. This research examines the socio-economic status of employees in the iron and steel industry, work fulfillment, visions of the industry, working circumstances, health facilities and their legal rights. The study is aimed at collecting empirical data on the subjects mentioned above, examining how these problems affect workers in the iron and steel sector and finding ways to resolve the sector's problems. Previous studies have been reviewed; the information of employees in the steel sector have been taken into account and empirical findings have been included. It proposes efficient steps to improve their socio-economic status and quality of life by providing better facilities / support for social welfare, and improving livelihoods in the area of the productivity and risk experienced by the staff of steel workers in its workplace and working setting. This study will provide a foundation for further research work in steel sector as no study has been done previously which address the socio-economic development of steel workers in the country. The methodology and model used in the study are new especially the use of Principal Component Analysis (PCA) and multinomial regression for such type of studies. The results are expected to assist the planners, policy makers and administrators involved revitalize the plan for safeguarding and protecting employees against the competitive pressure of contemporary steel manufacturing methods.

The rest of the study describes the research design, data collection and empirical framework, construction of the SES index, and description of explanatory variables in Section 2. Section 3 provides a description of the data and empirical research findings. Findings and recommendations are provided in Section 4.

## 2. Materials and methods

For this study purpose industries located in Badami Bagh area and along Sheikhupura road of Lahore were selected because of agglomeration which has been occurred in this area due to the proximity to M2 Section (Lahore-Islamabad) of Motorway. The city of Lahore is located in Pakistan's northeast portion. The district area ranged from 31˚-15' to 31˚-43'north latitude and 74˚-10' to 74˚ -39'east longitudes. Lahore City District has a total area of 1772 sq. km. [24].

### 2.1. Research design

This study is mainly planned as a 'primary research' which is based on 'survey design' for primary data collection from workers in the steel industry. A standardized questionnaire tool has

been used to collect data from these workers. For data collection, local contact persons were identified who facilitated the researchers in fixing time with management of industries to conduct interviews of workers.

To determine statistically reliable sample size, Cochran's sample size formula was used. The formula is presented in Eq (1) [25].

$$n = D \times \left[ \frac{Z^2 \times (p) \times (1-p)}{e^2} \right] \tag{1}$$

In the equation, sample size is denoted by $n$, $p$ represents the percentage of households picking a choice (expressed as decimal = 0.5), and $(p) \times (1-p)$ expresses an estimate of variance. $Z$ represents Z-value (1.96 for 95% confidence interval), $e$ is margin of error (0.08), and $D$ is design effect (1.50). By putting the values in the Eq (1), total sample size of 225 have been determined.

The study followed a stratified random sampling technique that means strata of industries were defined and after that workers were interviewed randomly among these strata. Design effect in our sample size formula is considered to compensate any loss of statistical robustness in the stratification procedure. The design effect is the ratio of the actual variance, under the sampling method actually used, to the variance computed under the assumption of simple random sampling. The sample size of 225 was approximately equally distributed across three clusters of industries.

The study was approved by the Qualification Examination Committee (an institutional Research Ethics Committee to ensure research quality and ethics) in the school before it began. During the data collection, the enumerators (first author and local contact persons) only interviewed workers aged between18-59 years. No minor was involved in any activity of this research. A verbal consent of the participants was obtained before starting the interview. The following statement was written on first page of questionnaire to satisfy the participants that this study is purely for academic purpose. "This is an academic study to fulfill the requirement of the Degree of 'Doctor of Philosophy' in Human Geography East China Normal University, Shanghai. All the information collected through this data collection tool will be kept highly confidential, and will be used purely for academic purpose. Respondents' identity, comments, suggestions and personal information will not be disclosed at any point of time. You are requested to participate in this important study."

## 2.2. Empirical methodology

This study used quintiles of socio-economic index, which is the proxy of household well-being and incomes, such as household assets, material resources and housing characteristics. This study used the socio-economic score index as a dependent variable and categorized into latent binary categories [26, 27]. This statistical approach is part of the logistic regression family and is a simple approach for generalizing binary variables [28, 29]. These quintiles categories are independence of observations and expressed as follows:

1. $\text{Prob}[y_{ij} = 1] = P_{i1}$ Household has socio-economic index score from 0–20% is the first quintile

2. $\text{Prob}[y_{ij} = 1] = P_{i2}$ Household has socio-economic index score from 20 to 40% is the second quintile

3. $\text{Prob}[y_{ij} = 1] = P_{i3}$ Household has socio-economic index score from 40 to 60% is the third quintile

4. $\text{Prob}[y_{ij} = 1] = P_{i4}$ Household has socio-economic index score from 60 to 80% is the fourth quintile

5. $\text{Prob}[y_{ij} = 1] = P_{i5}$ Household has socio-economic index score from 80 to 100% is the fifth quintile

The specification of the multinomial logit model is:

$$P_{ij} = \frac{\exp[\alpha_j + \beta_j X_j]}{\sum_j^k \exp[\alpha_j + \beta_j X_j]}$$

Where k is outcomes included in the model, which is precise the probability of household/individual to being in $j^{th}$ categories with characteristics $X_i$. The simplify probabilities of four outcomes are re-expressed as:

$$P_{i1} = \frac{\exp[\alpha_1 + \beta_1 X_i]}{1 + \exp[\alpha_1 + \beta_1 X_i] + \exp[a_3 + \beta_3 X_i] + \exp[\alpha_4 + \beta_4 X_i]}$$

$$P_{i2} = \frac{\exp[\alpha_2 + \beta_2 X_i]}{1 + \exp[\alpha_1 + \beta_1 X_i] + \exp[a_3 + \beta_3 X_i] + \exp[\alpha_4 + \beta_4 X_i]}$$

$$P_{i3} = \frac{\exp[\alpha_3 + \beta_3 X_i]}{1 + \exp[\alpha_1 + \beta_1 X_i] + \exp[a_3 + \beta_3 X_i] + \exp[\alpha_4 + \beta_4 X_i]}$$

$$P_{i4} = \frac{\exp[\alpha_4 + \beta_4 X_i]}{1 + \exp[\alpha_1 + \beta_1 X_i] + \exp[a_3 + \beta_3 X_i] + \exp[\alpha_4 + \beta_4 X_i]}$$

$$P_{i5} = \frac{1}{1 + \exp[\alpha_1 + \beta_1 X_i] + \exp[a_3 + \beta_3 X_i] + \exp[\alpha_4 + \beta_4 X_i]}$$

Since exp (0) = 1

Furthermore, log odds ratios can be expressed relative to any of the four categories under consideration.

**2.2.1. Construction of socioeconomic score bands for dependent variable.** This study used Principal Component Analysis (PCA) technique to construct the socioeconomic score (SES) index through the households' assets, and housing characteristics indicators, which is the proxy of household well-being. This technique (PCA) is commonly used in social science to measure socioeconomic status scores (SES), and reduces the number of variables in a dataset to less important dimension [30–33]. PCA has been validated as a method to describe SES differentiation within a population [34]. This (PCA) technique is used in two ways: covariance and correlation matrix techniques. However, this study used the correlation matrix technique to estimate the SES because the variables' measurement units are different. The mathematical

form of the PCA is as follows:

$$PC_1 = \alpha_{11}Y_1 + \alpha_{12}Y_2 + - - - + \alpha_1 nY_n$$

$$PC_1 = \alpha_{21}Y_1 + \alpha_{22}Y_2 + - - - + \alpha_2 nY_n$$

$$- - - - - - - - - - - - - - - - - --$$

$$- - - - - - - - - - - - - - - - - --$$

$$PC_1 = \alpha_{m1}Y_1 + \alpha_{m2}Y_2 + - - - + \alpha_m nY_n$$

Where PC1, PC2 and PCm are principal componenet equations with Yn different variables and mn equations weights. Formally, the linear combination of the SES for household i is calculated based on the following equation:

$$Y_i = \alpha_1\left(\frac{X_1 - \bar{X}_1}{S_1}\right) + \alpha_2\left(\frac{X_2 + \bar{X}_2}{S_2}\right) + \ldots\ldots\ldots + \alpha_K\left(\frac{X_K + \bar{X}_K}{S_K}\right)$$

Where yi is the SES, $X_K$ is the mean of assets and other social indicators; $S_K$ is the standard deviation of assets and other social indicators; and $\alpha_k$ are the weights for the assets and other social indicators.

Following the PCA calculation the first main component (PC1) is produced, with the highest weights between positive and negative values. To convert into standardized value, the average value of the score is subtracted from the actual value of the respective indicator and is divided by the standard deviation of score. Furthermore, PC1 is multiplied with these standardized scores to achieve the scores of each dimension. The summation of each dimension score gives the final SES at the household level. At the end the household level SESs are converted into five categories by using quintiles.

**2.2.2. Description of explanatory variables.** The selection of explanatory variables based on the Sustainable Livelihood Framework (SLF). These explanatory variables reflect the worker's endowments of the different forms of material and social well-being variables such as age, education, job status, skill level, job experience, dependency ratio, health insurance and transport facility etc. (Table 1).

## 3. Results and discussion

### 3.1. Descriptive analysis

This study explores the social development of workers in the steel industry in Pakistan. The deprivation of aspects of life as indicators of social development leads workers to social exclusion. Therefore, socially excluded individuals or groups cannot engage in social, economic and political activities contributing to stalled socioeconomic development at the public and person levels. As a result, social and economic growth is decreasing and criminal activities such as terrorism, street crime, abuse and community-level oppression and corruption are on the rise. Table 2 displays the descriptive statistics of the socioeconomic ranking. In addition, the SES score was divided into 5 quintiles, including the poorest (-1.489 to-0.822), the poor (-0.809 to-0.329), the middle (-0.302 to 0.162), the rich (0.190 to 0.968) and the richest (0.980 to 2.732).

Size of industry has an impact on the socioeconomic conditions (mainly determined by wages) of workers [35–37]. For men, quality-adjusted employer size wage effects are quite small and are mostly driven by lower wages for workers in the smallest firms (fewer than 20 workers) [38]. In our case, the situation is different, here, size of industry has negative impact (as shown in Fig 1) that means in larger production units, the socioeconomic status of workers is poor as compared to smaller production units and vice versa. This is because of as majority

**Table 1. Explanatory variables.**

| Explanatory variables | Description |
|---|---|
| Industry Type (dummy)* | Large = 1 and Small = 0 |
| Age (Years) | < = 20 (reference category) |
| | 21–30 |
| | 31–40 |
| | 41–50 |
| | 50> |
| Education (category) | Illiterate (reference category) |
| | Primary |
| | Middle |
| | Matric |
| | Intermediate or above |
| Skill Level (category) | Unskilled (reference category) |
| | Semi-skilled |
| | Skilled |
| Dependency (numeric) | Number of unemployed persons in HH |
| Job Status (dummy) | Permanent = 1 and Non-Permanent = 0 |
| Job Experience (numeric) | Number of years |
| Health Insurance (dummy) | Yes = 1 and No = 0 |
| Transport Facility (dummy) | Yes = 1 and No = 0 |

* Small scale industry: number of employees less than 50; Large scale industry: number of employees greater than 50.
Source: Authors.

of workers in larger units work on daily wage basis. On the other side, the workers in smaller production units enjoy contractual job status which leads them towards better socioeconomic status.

Skill level of workers also implies positive impact on socioeconomic score in labour market, there are evidences [39] which show that workers enriched with skills are in better position to bargain about salary as compared to those ones who are less skilled. having formal training increases the wage by 4.7% in the overall economy and the effect is highest in the primary sector [40]. This makes a clear difference in socioeconomic scores as it is reflected in our case in Fig 2 that skilled workers are at rich level as compared with unskilled or semiskilled workforce.

Job status of workers also play an important role in socioeconomic scoring. Generally, it is observed that the income earned by the regular workers is higher as compare to daily wage earners [41]. In our case (Fig 3), rich category of SES contains 18% employees each from permanent and non-permanent job status categories, while 37% of nonpermanent workers are

**Table 2. SES score categories.**

| Categories | Min | Max |
|---|---|---|
| Poorest (1st) | -1.489 | -0.822 |
| Poor (2nd) | -0.809 | -0.329 |
| Middle (3rd) | -0.302 | 0.162 |
| Rich (4th) | 0.190 | 0.968 |
| Richest (5th) | 0.980 | 2.732 |

Source: Authors' estimates.

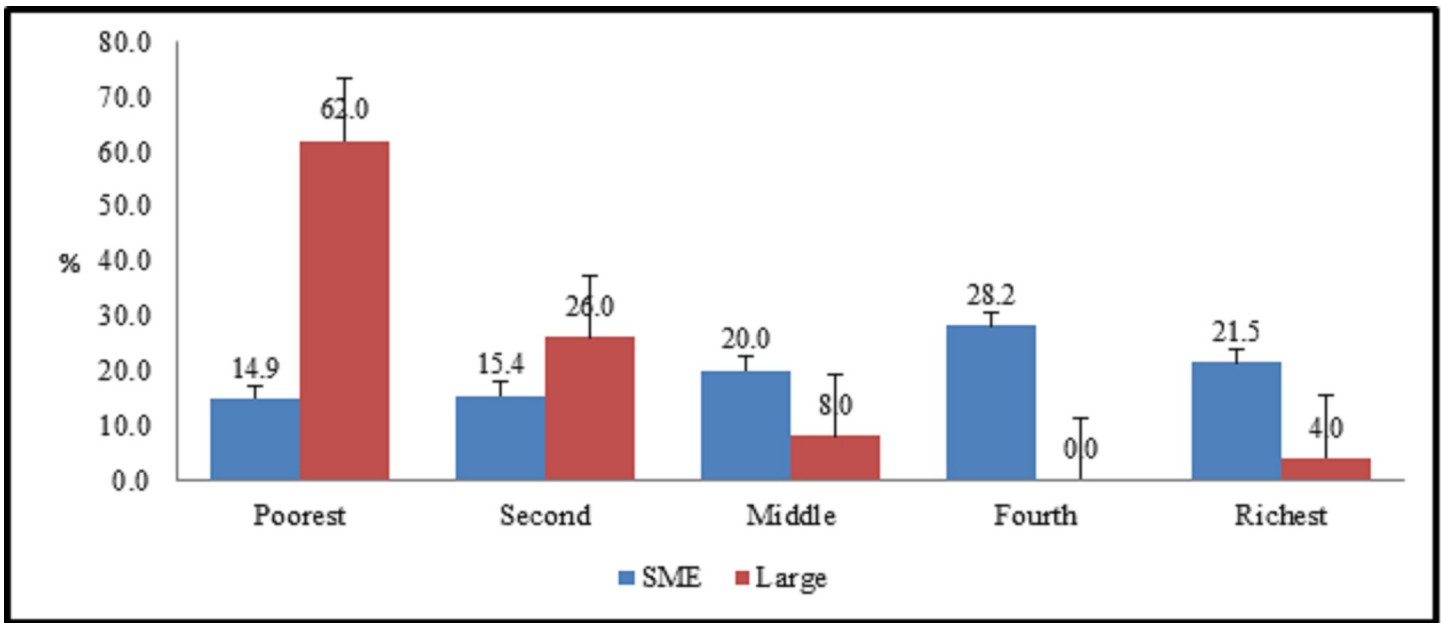

**Fig 1. Type of industry and socioeconomic.**

falling in poor SES category (poorest + second) as contrast to 45% in rich category (Richest + Fourth) of SES which is not a significant gap between these categories of SES. On the other side, in permanent category, 28% workers fall in rich SES (fourth + richest) category as compared to 55% in poor category (poorest + second) of SES. This behavior in this agglomeration of industry has some important reasons. Majority of these permanent workers are those who resides in their inherited houses located within surrounding buffers of industry. These workers have more than one earning sources, e.g. inherited agricultural land, part time jobs etc. which do not allow them to move away from their hometown. Furthermore, due to strong family

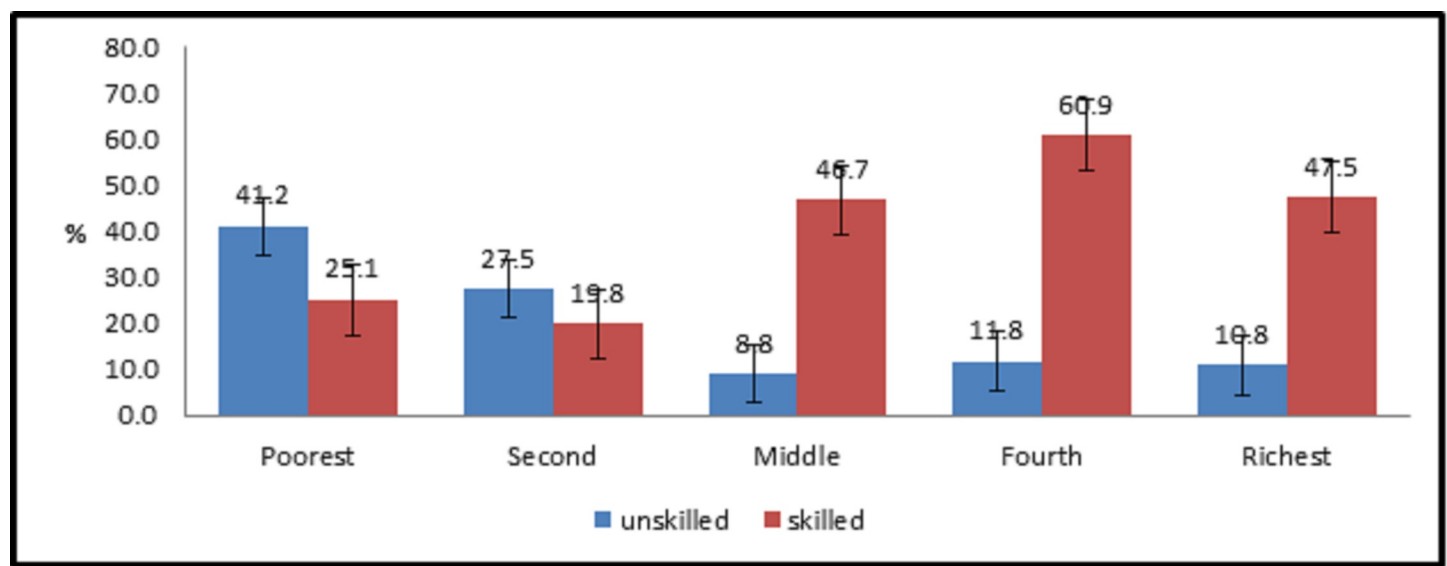

**Fig 2. Skill level and socioeconomic.**

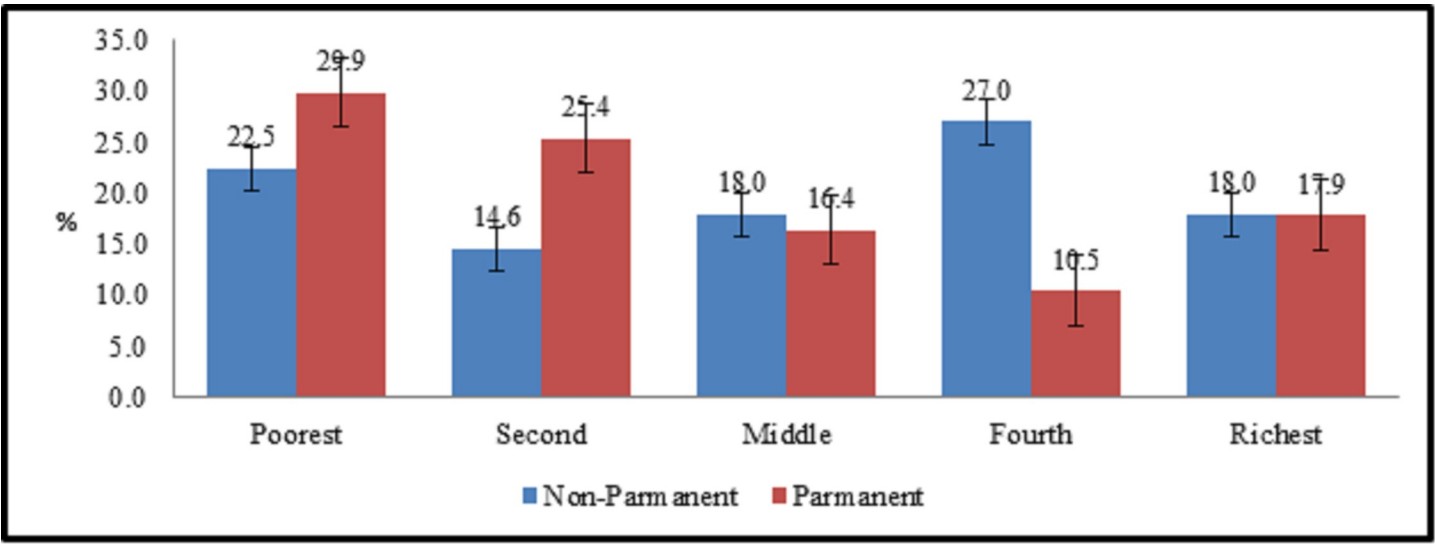

**Fig 3. Job status and socioeconomic.**

system and social bonding in society, they prefer to stay in their community. This prominent social and cultural trait of society becomes the main reason for industrialists to exploit these permanent workers as they know that these workers will never relocate and will work with low salaries.

Transport facility also affects the SES of workers. In this case (Fig 4), 41% of those who are not provided transport facility falls in poor (poorest + second) SES category as well as in rich (richest + fourth) SES category which means it is not as significant as other variables. On the other side, workers whom transport facility has been provided, about 57% falls in poor (poorest + second) SES category as compared to rich (richest + fourth) category which has 37%

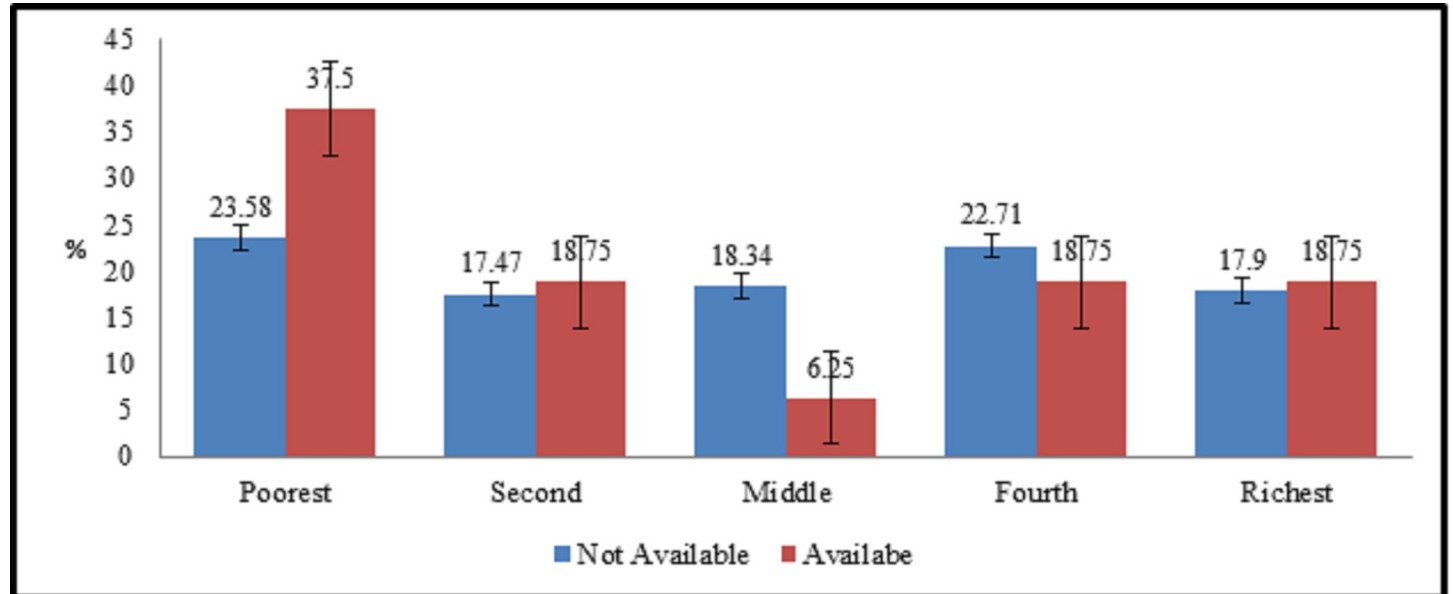

**Fig 4. Transport facility and socioeconomic.**

workers. These 37% are among those we have already discussed under permanent category having poor SES status because they reside outside the industrial units. The reason for this contrast is that the workers who do not use this facility have been provided residence inside the industry which saves their rent and transportation cost resulting their presence in rich SES category.

## 3.2. Empirical analysis

This research paper uses SES classifications as dependent variable and the industry type, age, education, skill level, number of dependents, job status, work experience, health insurance and transportation facilities are used as explanatory variables and are highly associated with SES. Table 3 shows multinomial logistic regression outcomes in four columns, the poorest, poor, middle and rich classes of SES, respectively. Category 5 SES (Richest) was used as a base outcome category. The socioeconomic aspects of the poorest, poor, middle and rich workers and their determinants are presented in Tables and illustrate the relationship between the explanatory variables and the SES likelihood. Understand the value of advantages and disadvantages across different groups. These kinds of decomposition analysis help formulate equitable policies. Social development is not simply a matter of physical resources. It is the reality of life that deprived person cannot maintain a proper standard of living. On the other hand, employment or income is not necessarily to end social exclusion between low and high-wage workers. Social exclusion exists due to market and infrastructure failure in industry [42, 43]. As a result

**Table 3. Empirical analysis.**

| Dependent Variable SES | | Poorest Category Coef. (1) | Poor Category Coef. (2) | Middle Category Coef. (3) | Rich Category Coef. (4) |
|---|---|---|---|---|---|
| **Base Outcome is Category (Richest)** | | | | | |
| **Industry Type** | Large = 1 and Small = 0 | 4.28* | 2.45* | 0.92** | -13.29** |
| **Age** | 21–30 | 0.83** | 1.09* | 0.85* | 1.66** |
| | 31–40 | 0.81* | 1.70* | 1.26* | 2.19** |
| | 41–50 | -1.97** | -2.10* | -1.20* | -2.60* |
| | 50> | -0.45* | -2.12** | -1.60* | -2.40* |
| **Education** | Primary | 1.04* | 0.26* | 0.83** | 0.14* |
| | Middle | 1.00** | 0.32* | 0.02* | 0.70*** |
| | Matric | 2.97** | 1.11* | 1.01* | 0.40** |
| | Intermediate or above | 1.47** | 0.81** | -1.26* | -0.10** |
| **Skill Level** | Semi-skilled | -3.61* | -2.57* | -0.01** | -0.56* |
| | Skilled | 6.01* | 5.21** | 0.35* | 0.53** |
| **Dependents** | | -0.19* | -0.04* | -0.13* | 0.02* |
| **Job Status** | Permanent = 1 and Non-Permanent = 0 | -0.40** | -0.21* | -0.06* | -1.69* |
| **Job Experience** | | 0.52* | 0.88** | 0.39** | 0.72** |
| **Health Insurance** | Yes = 1 and No = 0 | 0.36* | 0.78* | 0.02* | 0.48** |
| **Transport Facility** | Yes = 1 and No = 0 | 0.01* | 0.20* | -1.50* | -0.29* |
| **Constant** | | -0.55* | -2.09* | 1.60** | 2.53* |

*P<1%

**P<5% and

***P<10%, *Age (< = 20), Education (illiterate) and Skill level (Unskilled) used as reference categories.*

Source: Authors' estimates.

**Table 4. Relative risk ratio.**

| Dependent Variable SES | | Poorest Category Coef. (RRR) | Poor Category Coef. (RRR) | Middle Category Coef. (RRR) | Rich Category Coef. (RRR) |
|---|---|---|---|---|---|
| **Industry Type** | Large = 1 and Small = 0 | 72.46* | 11.59* | 2.51** | 0.00** |
| **Age** | 21–30 | 0.44* | 0.33* | 0.43* | 0.19* |
| | 31–40 | 0.45* | 0.18* | 0.29* | 0.11* |
| | 41–50 | 0.14** | 0.128* | 0.30* | 0.07* |
| | 50> | 0.64** | 0.12** | 0.20* | 0.09* |
| **Education** | Primary | 2.83* | 1. 30* | 2.29* | 0.87** |
| | Middle | 2.72* | 0.728 | 1.02* | 0.49* |
| | Matric | 19.57* | 3.04* | 2.75* | 1.50* |
| | Intermediate or above | 4.37* | 2.25** | 3.51* | 1.11* |
| **Skill Level** | Semi-skilled | 0.03** | 0.08** | 0.99** | 0.57* |
| | Skilled | 0.00*** | 0.01* | 0.70* | 0.59** |
| **Dependents** | | 0.83* | 0.96* | 1.14* | 1.03* |
| **Job Status** | Permanent = 1 and Non-Permanent = 0 | 0.67* | 0.81* | 1. 06** | 5.41** |
| **Job Experience** | | 1.69** | 2.42** | 1. 47* | 2.06** |
| **Health Insurance** | Yes = 1 and No = 0 | 1.44** | 2.18** | 1.02* | 1.61* |
| **Transport Facility** | Yes = 1 and No = 0 | 1.01* | 0.82* | 0.22** | 1.33** |

*P<1%

**P<5% and

***P<10%, *Age (< = 20), Education (illiterate) and Skill level (Unskilled) used as reference categories.*

Source: Authors' estimates.

of these inequities, this study used the socio-economic reasoning of their empirical findings. Table 4 indicates the ratio of the probability of choosing one outcome category over the probability of choosing the baseline category is often referred to as relative risk. Relative risk can be obtained by exponentiating the linear equations, yielding regression coefficients that are relative risk ratios for a unit change in the predictor variable. Results showed that the industry type (large) of sector is positively correlated with SES in the poorest, poor and bottom categories. While negatively associated with rich category and statistically significant. In large industries, employees are paid more than the small industries. In addition, large industries provide shelter, health and transportation facilities to employees [35–37]. The relative risk ratio indicates that one unit shift or improvement in the form of industry types from SES = 1 to 5 for the poorest is 72.46, the poor is 11.59, the middle is 2.51 and the wealthiest is 0.00 to be the richest SES. The age factor is strongly associated with SES at a certain era across the SES classifications and further decreased with a decreasing return to scale and negatively associated with SES across the SES classifications [44]. According to the outcomes of this research, 21–40 years of era have been seen as a successful career in the steel industry and have added positively to SES, which is the poorest to the wealthiest. In addition, older than 41 years of era have been noted as a declining and this is negatively associated with the poorest to the wealthiest classifications of SES. The primary reason behind this kind of relationship is the productivity and efficiency of employees, because we know that steel industry needs very strong labour to deal with heavy work. The relative risk ratio indicates that the estimated risk of staying poorest and poor in the age of 21–50 years is low compared to the reference category-the risk ratio for poorest

and poor is 0.44 and 0.33 for 21–30 years of age, 0.45 and 0.18 for 31–40 years of age and 0.14 and 0.128 for 41–50 years of age.

Education introduces the fundamental level of employee's abilities and the official education status of employees. Education is a key element of human resources and enables personal decision-making,leadership and signals of talent [45]. The use of education as a Socioeconomic Position (SEP) indicator has its historical origins in the status domain of Weberian theory, and it attempts to capture the knowledge related assets of a person [46]. In Pakistan, most sector employees experience regular difficulties in budgeting, operating hours and recording and reading medicine dosages and safety guidelines. This paper presents the formal education level (primary, middle, matric and intermediate or above) of workers, respectively. The low level (primary, middle and Matric) of education categories are positively correlated with SES but the high level (intermediate or above) is negatively correlated with middle and rich SES categories as comparison to base outcome category. Typically, the first line of emphasis is abilities, training and knowledge of specific skill sets in the sector of industries jobs, as discussed by [39, 40]. On the other hand Table 4 reported the relative risk ratiothat in low education the chances of becoming poorer to wealthier are higher as compared to higher education among steel industry workers. The relative risk ratio indicates 19.57 for the poorest, 3.04 for the poor, 2.75 for the middle, and 1.50 for the rich SES workers. Small and large business owners should attach the same importance to these skills. Furthermore, if they are lacking, current employees should be encouraged to build skills. The findings of this research also show that semi-skilled workers earn or pay less than skilled employees. Because of this, a semi-killed explanatory variable is the negative link to SES across categories and skilled variable is the positive link to SES across categories and statistically significant.

The Household dependency level is very broad with its associated socioeconomic and limited resources. This research finding indicate that in the poorest, poor and middle categories, household dependency is bad for SES, but great for rich categories. Household dependency is depleting resources, concrete as well as intangible, to the detriment of household members. The relative risk ratio (Table 4) shows that the estimated risk of remaining poorest and poorer is high because the estimated risk is lower value shift as one unit increases in dependents. Although [41, 47] concluded that permanent employees receive more salary than non-permanent employees and are much better in SES and [48] discussed that temporary workers are being exploited by the contractors. But here in our case, permanent job status is negatively related with SES across the categories because majority of permanent workers do not relocate, hence exploited by employers in steel industries as well as other industries. Job experience is positively correlated with SES across the categories. The reason is that less educated workers start working in very young age due to poor economic conditions hence do not carry on with their education. Carrying more experience, they are normally permanent, more skilled and more attractive for large industries. Combining these qualities with high salaries, they fall under rich category of SES [45]. The relative risk ratio indicates that the approximate risk of being the poorest to the wealthiest is 1.69 for the poorest, 2.42 for the poor in work experience, 1.44 for the poorest and 2.18 for the poor in health insurance and 1.01 for the poorest in transport facilities.

Previous research [45] concluded that money is not everything. Staff enjoy more than just compensation. Employees are also concerned with non-wage benefits, such as insurance and pension benefits. The selection of these benefit programs varies among employees: older workers are more likely than younger to be concerned with pension benefits or better health insurance. Same is reflected in our case where health insurance treatment is positive related with SES which means that where health treatment is provided, socioeconomic condition of workers is better than those where it is not provided. The transport facility has a favorable

relationship with the poorest and poor SES and an adverse relationship with the middle and wealthy SES. Workers coming from outside the factory, especially the poorest and poor SES groups, cannot afford daily transportation costs, and if it is provided by the employers, they can save traveling expenses to improve their socio-economic status. This situation is not only associated with steel industry, it is the common problem of workers of all sectors [49].

## 4. Conclusions and recommendations

Social Development Inequalities can be measured by a number of social and economic factors, some of those have been raised and discussed in this study. It is a relevant social status scoring of workers of a particular industry rather than an absolute generalization for all industries. Four categories of socioeconomic index such as less than or equal to 25%, 50%, 75% and more than 75% have been used which is the reflection of household well-being. The aspects of inequality have been highlighted by using the following indicators: material resources, job status and experience, skill and education level, provision of health insurance and transport facility, number of dependents etc.

Based on the results of this research, 21 to 40 years of age were considered a successful career in the steel industry and contributed positively to SES. Moreover, over 41 years of age have been rated as a decline and this is negatively associated with the poorest of the richest classifications of SES. The relative risk ratio indicates that the estimated risk of remaining poor from 21 to 50 years of age is low relative to the reference category. The risk ratio for poorest and poor is 0.44 and 0.33 for 21–30 years of age, 0.45 and 0.18 for 31–40 years of age and 0.14 and 0.128 for 41–50 years of age. Education introduces the basic level of skills of employees and the official educational status of employees. In Pakistan, most employees in the sector regularly find it difficult to budget, stick to business hours, and record and read medication doses and safety guidelines. The low level of education of the categories is positively correlated with the SES, but the high level is negatively correlated with the medium and rich SES categories in comparison with the baseline outcome category. These findings indicated that the high level of formal education is less important than technical skills and education in industrial sectors.

Owners of both small and large companies should consider these skills equally important. As well, if inadequate, current employees should be encouraged to develop skills. The results of this study also show that semi-skilled workers earn or pay lower wages than skilled workers. For this reason, a semi-skilled explanatory variable is the negative link to SES between categories and the skilled variable is the positive link to SES between categories and statistically significant. The level of household dependence is very wide, with limited socioeconomic resources associated with it. This research shows that in poorest, poor and medium-sized categories, household dependence is more vital for the SES improvement. The dependence of households depletes resources, both concrete and intangible, to the detriment of household members. The relative risk ratio shows that the estimated risk of remaining poor is high because the estimated risk is a lower value change as one unit increases in dependents.

Permanent job status is negatively related with SES across the categories because majority of permanent workers do not relocate, hence exploited by employers in steel industries as well as other industries. This is because less educated workers start working very young due to poor economic conditions and therefore do not continue their education. With more experience, they are usually permanent, more qualified and more attractive to major industries. Combining these attributes with high wages, they fall into the rich category of SES.

Employees also have an interest in non-salary benefits, such as insurance and pension benefits. The choice of these benefit programs varies by employee: older workers are more likely than younger workers to be interested in retirement benefits or better medication. It is the

same in our case where health insurance treatment is positively related to SES, which means that when health treatment is provided, the socio-economic situation of workers is better than that of workers who do not benefit from such support. The ease of transportation has a favorable relationship with the poor SES and a negative relationship with rich SES. Workers from outside the factory, in particular the poor SES groups, cannot afford daily transportation costs, and if provided by employers, they may save travel costs to enhance their socio-economic status.

Based on the findings of this study, it is suggested that focusing on problems e.g. safety at job environment, health insurance, compliance of labour laws, occasional bonuses and annual increments etc. can play an important role to improve the level of socioeconomic status score and to decrease the gaps regarding social development inequalities among workers of steel industry in the country.

## Supporting information

**S1 File. Questionnaire socio-economic conditions of steel industry workers.**
(DOCX)

## Acknowledgments

The authors express their gratitude to the editor and two anonymous reviewers for their supportive and informative comments, which greatly improved the article. We would also like to express our gratitude to all of the factory owners who allowed us access and the factory workers who agreed to be interviewed. Despite the fact that they remain anonymous, this study would not have been possible without their participation.

## Author Contributions

**Conceptualization:** Shahid Karim.

**Data curation:** Shahid Karim, Abdul Hameed.

**Formal analysis:** Shahid Karim, Kong Xiang, Abdul Hameed.

**Investigation:** Shahid Karim.

**Methodology:** Shahid Karim, Kong Xiang.

**Project administration:** Kong Xiang.

**Resources:** Kong Xiang.

**Software:** Shahid Karim, Abdul Hameed.

**Supervision:** Kong Xiang.

**Validation:** Shahid Karim, Kong Xiang, Abdul Hameed.

**Visualization:** Shahid Karim, Kong Xiang, Abdul Hameed.

**Writing – original draft:** Shahid Karim.

**Writing – review & editing:** Shahid Karim, Kong Xiang.

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
