## [Decision Letter · Decision Letter 0]

30 Mar 2021

PONE-D-21-05693

Investigating social development inequality among steel industry workers in P akistan:

 A contribution to social development policies

PLOS ONE

Dear Dr. Karim,

Thank you for submitting your manuscript to PLOS ONE. After careful consideration, we feel that it has merit but does not fully meet PLOS ONE’s publication criteria as it currently stands. Therefore, we invite you to submit a revised version of the manuscript that addresses the points raised during the review process.

The manuscript investigates an interesting topic. However, it needs significant improvements in introduction, methodology, results and conclusions section.

We look forward to receiving your revised manuscript.

Kind regards,

Abid Hussain

Academic Editor

PLOS ONE

Journal Requirements:

2. Please include additional information regarding the survey or questionnaire used in the study and ensure that you have provided sufficient details that others could replicate the analyses. For instance, if you developed a questionnaire as part of this study and it is not under a copyright more restrictive than CC-BY, please include a copy, in both the original language and English, as Supporting Information."

3. Thank you for including your ethics statement: 

"The study was approved by the Research Ethics Committee (an institutional body to ensure research quality and ethics) in the school before it began. During the data collection, the enumerators (first author and local contact persons) only interviewed workers aged between18-59 years. No minor was involved in any activity of this research. A verbal consent of the participants was obtained before starting the interview. The following statement was written on first page of Questinnaire to satisfy the participants that this study is purely for academic purpose.

“This is an academic study to fulfill the requirement of the Degree of ‘Doctor of Philosophy’ in Human Geography East China Normal University, Shanghai. All the information collected through this data collection tool will be kept highly confidential, and will be used purely for academic purpose. Respondents’ identity, comments, suggestions and personal information will not be disclosed at any point of time. You are requested to participate in this important study.” ".   

Once you have amended this statement in the Methods section of the manuscript, please add the same text to the “Ethics Statement” field of the submission form (via “Edit Submission”).

4. Thank you for stating the following in the Financial Disclosure section:

We note that one or more of the authors are employed by a commercial company: "Innovative Development Strategies"

5. We note you have included a table to which you do not refer in the text of your manuscript. Please ensure that you refer to Table 3 in your text; if accepted, production will need this reference to link the reader to the Table.

Additional Editor Comments (if provided):

Dear authors,

Thank you for submitting a very interesting article. The current version of article needs significant revisions particularity in Introduction, methodology and conclusions sections. I also urge to improve discussion of results.

Reviewers' comments:

Reviewer's Responses to Questions

**Comments to the Author**

1. Is the manuscript technically sound, and do the data support the conclusions?

Reviewer #1: Yes

Reviewer #2: Yes

2. Has the statistical analysis been performed appropriately and rigorously? 

Reviewer #1: Yes

Reviewer #2: Yes

3. Have the authors made all data underlying the findings in their manuscript fully available?

Reviewer #1: Yes

Reviewer #2: Yes

4. Is the manuscript presented in an intelligible fashion and written in standard English?

Reviewer #1: Yes

Reviewer #2: Yes

5. Review Comments to the Author

Reviewer #1: Title: Investigating social development inequality among steel industry workers in Pakistan: A contribution to social development policies

Manuscript ID: PONE-D-21-05693

Referee Recommendation: MINOR REVISIONS

I appreciate the author(s) for studying a sensitive issue of social inequality among workers of steel industry in Pakistan and policy formulations for betterment of targeted population. However, I have noted some issues which may render further quality to the paper, as following:

Title: The title is ok.

Abstract: The abstract has been nicely composed. However, there must be few lines on data collection tool and exact studied population i.e., name of development zone.

Introduction: This section has been nicely drafted. However, I would like to recommend the author(s) to update this section via citing recent literature along with consistent formatting for citing these studies. Secondly, the section can be further improved grammatically via good academic writing.

Study objectives and significance: This section is nicely written, just need to follow the practice of good academic writing through rephrasing few sentences of this section.

Material and Methods: The research design, data collection technique and sampling method are appropriate, and the section has been nicely elaborated. There is a just need to check for formatting and grammatical mistakes.

Empirical analysis: The author has nicely explained the results along with appropriate model selection. I am just wondering how did the author check the independence of four categories of dependent variable!!

Conclusions and recommendations: This section has been nicely drafted.

Reviewer #2: The paper studies the social development inequalities among steel industry workers in Pakistan using data from 225 workers. The authors use Principal Component Analysis to construct the socioeconomic score (SES) index and employ multinomial logistic regression model to four categories of socioeconomic index. While the topic is of interest, the authors may follow the following recommendations:

1. Make the abstract more concrete and informative about key findings.

2. Introduction section needs substantial revisions to present the research problem, research gap and the contribution of this study. The author(s) should carry out a thorough literature survey of papers published in a range of top journals in the last three/four years relating to the topic addressed in your manuscript.

3. Methodology Section talks about the methodology and attributes of the data collection and sample size. The important component, which is neglected here is the justification of the chosen variables.

4. The authors used four categories of socioeconomic index, which they claim, are independent to each other. Please justify how are they independent?

5. Their definitions of four probabilities are also not consistent with each other.

6. The authors need to cite references in their Empirical Methodology.

7. The authors present their socioeconomic score against two types of industrial units i.e. SME and large in Figure 1 but do not provide their classification criteria.

8. The authors need to cite sources of figures and tables.

9. The authors need to provide proper socio-economic reasoning of their empirical results.

10. The conclusions of this study are very general in nature. If so then such conclusion will not make any sense. Make the section more concrete with viable suggestions for policy.

6. PLOS authors have the option to publish the peer review history of their article (what does this mean?). If published, this will include your full peer review and any attached files.

Reviewer #1: No

Reviewer #2: No

---

## [Author Response · Author response to Decision Letter 0]

8 May 2021

All the comments and suggestions from the editor and anonymous reviewers have been incorporated and described in the document "Response to reviewers".

---

## [Decision Letter · Decision Letter 1]

24 May 2021

PONE-D-21-05693R1

Investigating social development inequality among steel industry workers in Pakistan:

 A contribution to social development policies

PLOS ONE

Dear Dr. Karim,

Thank you for submitting your manuscript to PLOS ONE. After careful consideration, we feel that it has merit but does not fully meet PLOS ONE’s publication criteria as it currently stands. Therefore, we invite you to submit a revised version of the manuscript that addresses the points raised during the review process.

The article has improved a lot after revisions. However, I still have couple of minor points which need your attention. For multi-category variables - age, education and skill level - in the regression model, please present reference (base) categories in Tables 2-4. Moreover, improve the quality of figures, and label their Y-axis adequately. 

We look forward to receiving your revised manuscript.

Kind regards,

Abid Hussain

Academic Editor

PLOS ONE

Journal Requirements:

Additional Editor Comments (if provided):

Dear Author(s),

Thank you so much adequately addressing the comments from two reviewers. I still have one minor point which needs clarification.

In the regression model, three variables - age, education and skill level - have multiple discrete categories but their reference (base) categories are not highlighted neither in Table 2 (on variables description), nor in tables 3-4 on results. I suggest to present reference categories in Table 2, and provide note below Tables 3-4 to highlight reference categories.

Please also provide good quality graphs/figures. Current graphs are little blur.

Reviewers' comments:

Reviewer's Responses to Questions

**Comments to the Author**

1. If the authors have adequately addressed your comments raised in a previous round of review and you feel that this manuscript is now acceptable for publication, you may indicate that here to bypass the “Comments to the Author” section, enter your conflict of interest statement in the “Confidential to Editor” section, and submit your "Accept" recommendation.

Reviewer #1: All comments have been addressed

Reviewer #2: (No Response)

2. Is the manuscript technically sound, and do the data support the conclusions?

Reviewer #1: Yes

Reviewer #2: Yes

3. Has the statistical analysis been performed appropriately and rigorously? 

Reviewer #1: Yes

Reviewer #2: (No Response)

4. Have the authors made all data underlying the findings in their manuscript fully available?

Reviewer #1: Yes

Reviewer #2: (No Response)

5. Is the manuscript presented in an intelligible fashion and written in standard English?

Reviewer #1: Yes

Reviewer #2: (No Response)

6. Review Comments to the Author

Reviewer #1: I really appreciate the way authors have improved this manuscript. Therefore, I would like to thank for addressing my comments.

Reviewer #2: (No Response)

7. PLOS authors have the option to publish the peer review history of their article (what does this mean?). If published, this will include your full peer review and any attached files.

Reviewer #1: No

Reviewer #2: No

---

## [Author Response · Author response to Decision Letter 1]

25 May 2021

Response to editor: 

We ensure that our manuscript meets PLOS ONE's style requirements

A copy of Questionnaire form is attached 

Ethics statement is revised in the Methods section of the manuscript with full name of ethics committee/institutional review board(s) that approved our study

An amended Funding Statement is provided in the revised cover letter 

An updated Competing Interests Statement is provided in the revised cover letter

Table 3 incorporated 

Base (Reference) categories of Age, Education and Skill Level are incorporated in revised manuscript in Table 2 at page number 9 and 10 and below table 3 and 4 (as note) on page 14 and 16

Figures have been edited in good quality and uploaded as per requirements of PLOS ONE

Response to reviewer #1 

Thanks, we acknowledge and appreciate your comment

Incorporated as per suggestion

Incorporated as per suggestion

Sentences have been rephrased to shape the document more professional and academic 

Formatting and grammatical mistakes are checked once again and possible correction have been made

This study used quintiles of socio-economic index, which is the proxy of household well-being and incomes, such as household assets, material resources and housing characteristics. This statistical approach is part of the logistic regression family and is a simple approach for generalizing binary variables. These quintiles categories are independence of observations. Methodological section rephrased according to the comment. 

Thanks, we acknowledge and appreciate your comment 

Response to Reviewer #2

Incorporated as per suggestion

Incorporated as per suggestion

Incorporated under heading of “description of explanatory variables”. 

The selection of explanatory variables based on the Sustainable Livelihood Framework (SLF). These explanatory variables reflect the worker’s endowments of the different forms of material and social well-being variables such as age, education, job status, skill level, job experience, dependency ratio, health insurance and transport facility etc

This study used quintiles of socio-economic index, which is the proxy of household well-being and incomes, such as household assets, material resources and housing characteristics. This statistical approach is part of the logistic regression family and is a simple approach for generalizing binary variables. These quintiles categories are independence of observations. Methodological section rephrased according to the comment. 

Incorporated as per suggestion

Incorporated as per suggestion

Classification criteria of industry has addressed now in table under heading “description of explanatory variables”

Incorporated as per suggestion

Understand the value of advantages and disadvantages across different groups. These kinds of decomposition analysis help formulate equitable policies. Social development is not simply a matter of physical resources. It is the reality of life that deprived person cannot maintain a proper standard of living. On the other hand, employment or income is not necessarily to end social exclusion between low and high-wage workers. Social exclusion exists due to market and infrastructure failure in industry (Saith, 2001) (Hameed & Qasir, 2019). As a result of these inequities, this study used the socio-economic reasoning of their empirical findings.

Conclusion findings have been rewritten as per suggestion.

---

## [Editor Report · Decision Letter 2]

28 May 2021

Investigating social development inequality among steel industry workers in Pakistan:

 A contribution to social development policies

PONE-D-21-05693R2

Dear Dr. Karim,

We’re pleased to inform you that your manuscript has been judged scientifically suitable for publication and will be formally accepted for publication once it meets all outstanding technical requirements.

Kind regards,

Abid Hussain

Academic Editor

PLOS ONE
---

## [Editor Report · Acceptance letter]

1 Jun 2021

PONE-D-21-05693R2 

Investigating social development inequality among steel industry workers in Pakistan:
 A contribution to social development policies 

Dear Dr. Karim:

I'm pleased to inform you that your manuscript has been deemed suitable for publication in PLOS ONE. Congratulations! Your manuscript is now with our production department. 

Kind regards, 

on behalf of

Dr. Abid Hussain 

Academic Editor

PLOS ONE